# Towards More Efficient Rényi Entropy Estimation

**DOI:** 10.3390/e25020185

**Published:** 2023-01-17

**Authors:** Maciej Skorski

**Affiliations:** Faculty of Mathematics, Informatics and Mechanics, University of Warsaw, 00-927 Warszawa, Poland; maciej.skorski@mimuw.edu.pl

**Keywords:** Rényi entropy, adaptive estimation, collision estimation, birthday paradox

## Abstract

Estimation of Rényi entropy is of fundamental importance to many applications in cryptography, statistical inference, and machine learning. This paper aims to improve the existing estimators with regard to: (a) the sample size, (b) the estimator adaptiveness, and (c) the simplicity of the analyses. The contribution is a novel analysis of the generalized “birthday paradox” collision estimator. The analysis is simpler than in prior works, gives clear formulas, and strengthens existing bounds. The improved bounds are used to develop an adaptive estimation technique that outperforms previous methods, particularly in regimes of low or moderate entropy. Last but not least, to demonstrate that the developed techniques are of broader interest, a number of applications concerning theoretical and practical properties of “birthday estimators” are discussed.

## 1. Introduction

### 1.1. Motivation and Background

The aim of *entropy estimation* is to approximately compute the Rényi entropy of an *unknown probability* distribution using only *observed samples*. Since Rényi entropy is the most established and popular uncertainty measure, the problem is not only one of fundamental interest to information theory [1], but also one of importance to a number of applied research areas. These applications of Rényi entropy include, in particular, quantifying diversity in ecology [2,3,4], statistical mechanics [5,6], thermodynamics [7], characterizing properties of probability distributions [8,9], DNA sequencing [10,11], network anomaly detection [12,13], clustering [14,15], data mining [16,17], predictive modelling [18], as well as security and cryptography [19,20,21,22,23,24,25,26,27,28].

To state the problem formally, if we consider a fixed discrete distribution with probability mass function $p_{X}$, the Rényi entropy [1] of some fixed positive order *d* is defined as
(1)$$\begin{array}{c} H_{d}\left(p_{X}\right)≜\frac{1}{1-d}\log_{2}\left(\sum_{x}p_{X}{\left(x\right)}^{d}\right). \end{array}$$The challenge is to estimate this quantity from *n* independent samples $X_{1},\ldots ,X_{n}{∼}^{iid}p_{X}$. More precisely, we seek an *explicit function of samples*
$\hat{H}$ such that the approximation
(2)$$\begin{array}{c} \hat{H}\left(X_{1},\ldots ,X_{n}\right)\approx H_{d}\left(p_{X}\right) \end{array}$$
holds with a small error, high probability, and possibly minimal sample size *n*. We are interested in *non-parametric estimation*, as the distribution *X* remains unknown.

As in prior work on Rényi entropy estimation, we focus on *integer orders*. This is not limiting, because Rényi entropies of positive integer orders: (a) encode the complete information about the distribution [29], (b) are sufficient for practical applications due to known smoothing and interpolation properties [30,31], and finally (c) are more efficient to estimate from the algorithmic perspective [32].

### 1.2. Related Work

The most natural, albeit not most efficient, are so-called *plugin estimators*, which insert a non-parametric estimate of the probability mass function into the entropy formula [33]. As opposed to that, sample-optimal estimators for Rényi entropy are more involved, as shown in a relatively recent line of work [32,34,35]. Loosely speaking, these estimators relate the entropy to *collision probabilities* and then take advantage of the *birthday paradox*. More specifically, base estimations are obtained by counting collisions among tuples in the observed sample, and then are optionally run in parallel to boost the statistical confidence. The birthday paradox intuitively explains why the resulting algorithms are *sublinear in the alphabet size*. Specifically, for the alphabet of size *K*, the sample-optimal estimation takes $O_{d}\left(K^{1-1/d}\right)$ samples (the asymptotic notation $O_{d}\left(\right)$ hides dependencies on *d*.) to achieve an additive error of at most 1 and a level of confidence of at least 2/3 [32,34,35]. The aforesaid state-of-the-art estimators are *asymptotically minimax optimal*; that is, they achieve the asymptotically minimal sample size over the worst choice of the sampling distribution. However, their known analyses leave room for improvement with regard to simplicity, numerical precision, estimation adaptiveness, and techniques used. We elaborate on these issues below:1.**Lack of simplicity and numerical precision.** The analyses of the state-of-the-art estimators [32,34,35] struggle with analysing the variance of collision estimators, which is tackled either by *poissonization approximations* [32,34] (which carry their own overhead) or by using *involved combinatorics* [35]. As a consequence, the variance bounds are available in asymptotic “big-Oh” notation hiding constants and higher-order dependencies (such as relations to the order *d*), and are not suitable for applications in statistics or cryptography, which demanding precise formulas. This point was already raised in the context of applied works on physically unclonable functions [36].2.**Adaptiveness gap.** The focus of prior research was on establishing bounds under the *worst-case* choice over all distributions [32,34]. This is overly pessimistic because distributions that arise in practical applications are of a different structure than those occurring in this worst-case scenario analysis (the worst-case choice is known to be a mixture of uniform and Dirac distributions). The bounds were somewhat improved in the follow-up work [35] where a *prior entropy bound* is assumed. Still, there is a gap as the entropy bound is usually not known prior to the actual experiment. In fact, getting an entropy bound might be more costly than its application.3.**Lack of established techniques.** Prior work focused on delivering asymptotic formulas and did not elaborate much on techniques that could help obtain simpler and tighter bounds. The difficulty of analyzing collision estimators is a recurring issue, well-known to the researchers working on property testing [37]. Do we have a systematic method of handling it?

### 1.3. Our Contribution

This work fills the aforementioned gaps with the following contributions:1.**Simpler and more accurate analysis** of collision estimators, the main building blocks of the state-of-the-art Rényi entropy estimators. We analyze the collision estimators as *kernel averages* with the technique of *Hoeffding’s decomposition*. This novelty brings the promised simplicity and improvement in accuracy.2.**Adaptive estimation of Rényi entropy**, using no prior knowledge of the sampling distribution. The sample size cost of the presented algorithm is essentially optimal (up to a poly-logarithmic factor).3**Modular approach** using the established methods of *U-statistics* and *Robust Mean Estimation*. Specifically, we point out that the moment estimation problem, which Rényi entropy estimation reduces to, can be seen as the estimation of certain *U-statistics*. While the dedicated statistical theory provides the bias–variance analysis, the confidence can be independently boosted by techniques of *Robust Mean Estimation*.

This paper aims to solve the two mentioned bottlenecks and, in this way, to close the gap between the theory-oriented state-of-the-art and the demand coming from applied researchers and their practical use cases, such as [36].

### 1.4. Organization

The notation and preliminary concepts are discussed in Section 2. The technical results and applications are presented in Section 3. The proofs are discussed in Appendix A, and the work is concluded in Section 4.

## 2. Preliminaries

### 2.1. Basic Notation

Throughout the paper, $p_{X}$ is the probability mass function of a fixed discrete distribution over an alphabet of size *K* and $X_{1},\ldots ,X_{n}$ are observed independent samples from $p_{X}$. We denote [n]={1,…,n} and let [n]d denote the collection of all *d*-element subsets of *n*.

### 2.2. Estimation of Entropy, Moments, and Collisions

We leverage the following observation from prior work: the task of Rényi entropy estimation is equivalent to the task of *moment estimation*. More precisely, the *d*-th moment of the probability mass function $p_{X}$ is defined as
(3)$$\begin{array}{c} P_{d}≜\sum_{x}p_{X}{\left(x\right)}^{d}, \end{array}$$
and then—immediately from definition—we have the following result:

**Proposition** **1.***An estimate $\hat{H}$ of the Rényi entropy of order d (defined in (Equation 1)) has an additive error of ϵ if and only if $2^{\hat{H}}$ is an estimate of the d-th moment (defined in (Equation 3)) with a* relative error *of $\epsilon^{\prime }=2^{\epsilon (d-1)}-1$.*

Solving the (equivalent) problem of moment estimation is more convenient due to the beautiful representation of moments as collision probabilities. More precisely, we have
(4)$$\begin{array}{c} P_{d}=P\left(X_{1}=X_{2}=\ldots =X_{d}\right). \end{array}$$

### 2.3. U-Statistics

For a symmetric real function *h* of *d* arguments, the *U-statistic* with kernel *h* of the sample $X_{1},\ldots ,X_{n}$ is defined as:(5)Uh(X1,…,Xn)≜nd−1∑1⩽i1<…<id⩽nh(Xi1,…,Xid).The *U-statistic* gives an unbiased estimate of the function expectation, hence its name. *U-statistics* were invented by Hoeffding [38] to extend certain results, such as concentration bounds, to sums of partly dependent terms. Many statistical quantities can be related to *U-statistics*; for example, moments or sample variances [39]. In the same spirit, we will see estimators of the collision probability in Equation (Equation 3) as *U-statistics* and use those to establish their desired properties.

### 2.4. Robust Mean Estimation

It is difficult to directly obtain high confidence bounds (such as those of the Chernoff–Hoeffding type) for moment estimators. Instead, we will boost weaker bounds obtained from bias–variance analyses. To this end, we combine independent runs of estimators into high-confidence bounds using the technique of *Robust Mean Estimation*.

Estimating the mean of a distribution from i.i.d. samples is not trivial: the “obvious” use of the empirical mean is inaccurate for heavy detailed distributions. Following the recent survey [40], we mention here two solutions:the *median-of-means* approach organizes data (such as independent algorithm outputs) into blocks and computes the median of means within blocks.the *trimmed mean* approach takes the mean of independent runs, excluding a certain fraction of smallest and biggest outcomes (removing outliers).

We note that any robust mean estimation can be used to achieve confidence boosting. In this work, we stick to the median-of-means. The following result discusses its performance.

**Proposition** **2**(Performance of Median-of-Means [40]). *Let $Z_{1},\ldots ,Z_{n}$ be i.i.d. random variables with mean μ and variance $\sigma^{2}$. For k=⌈8log(1/δ)⌉, split $Z_{1},\ldots ,Z_{n}$ into k blocks and let $\hat{\mu }$ be the median of the means within blocks. Then, with probability 1−δ,*
(6)$$\begin{array}{c} |\hat{\mu }-\mu |⩽\sigma \sqrt{\frac{4}{\left\lfloor n/k\right\rfloor }}. \end{array}$$

### 2.5. Moment Bounds

We will need some bounds on moments of probability distributions, in order to simplify formulas that arise from variance analysis. Specifically, we will use these auxiliary results to express higher-order moments in terms of moments of small order (d=2 and d=3).

**Proposition** **3.**
*For any probability distribution $p=(p_{i})$, we have*

$$\begin{array}{c} \sum_{i}p_{i}^{2d-k}⩾{\left(\sum_{i}p_{i}^{d}\right)}^{2} \end{array}$$

*for any integer k,d such that 1⩽k⩽d. Moreover, with $p_{\max}≜\max_{i}p_{i}$ it holds that*

$$\begin{array}{c} \sum_{i}p_{i}^{2d-k}⩾{\left(\sum_{i}p_{i}^{d}\right)}^{2}/p_{\max}^{k-1}. \end{array}$$



**Proposition** **4.**
*For any non-negative sequence $\left(p_{i}\right)$, it holds that the quantity ${∥p∥}_{d}≜{\left(\sum_{i}p_{i}^{d}\right)}^{1/d}$ decreases in d⩾1.*


## 3. Results

Following the convention from prior work, our results are stated for *moment estimation*, which is equivalent to entropy estimation as discussed in Proposition 1. Throughout the rest of the paper, we keep this reduction in mind.

### 3.1. Simpler & More Accurate Moment Estimation

The first novelty offered in the current work is a *simplified and strengthened* variance analysis of the state-of-the-art moment estimator, presented in Algorithm 1. Differently than in prior work, we write the estimator output as a *kernel average* of the function $h\left(x_{i_{1}},\ldots ,x_{i_{d}}\right)≜I\left(x_{i_{1}}=\ldots x_{i_{d}}\right)$ over the *d*-element subsets of the sample. This approach is not only more readable but ultimately also more accurate, as it links the task of moment estimation to the established theory of *U-statistics* [38].

On top of that comes the *refined high-confidence moment estimator* in Algorithm 2, which we build on robust mean estimators [40]. Due to this modularity, it uses fewer samples than the direct approach from prior work [34].

**Algorithm 1:** BIRTHDAY MOMENT ESTIMATOR**Data**:
entropy order *d*samples $X_{1},\ldots ,X_{n}$, n⩾d, from an unknown distribution with p.m.f. $p_{X}$
**Result**: an estimate ${\hat{P}}_{d}$ of $P_{d}=\sum_{x}p_{X}{\left(x\right)}^{d}$, the *d*-th moment of $p_{X}$

C←#{i1…id}∈[n]d:Xi1=Xi2=…=Xid



P^d←C/nd

**return** 
${\hat{P}}_{d}$

**Theorem** **1**(Bias–Variance Analysis of Moment Estimator). *With the notation as above, the output of Algorithm 1 is unbiased:*



(7)
$$\begin{array}{c} E[{\hat{P}}_{d}-P_{d}]=0, \end{array}$$




*and its variance equals*

(8)
Var[P^d]=∑k=1ddkn−dd−k(P2d−k−Pd2)nd.




*In particular, for any n⩾d and ϵ>0 we can upper-bound the variance as*

(9)
$$\begin{array}{c} \mathrm{Var}\left[{\hat{P}}_{d}\right]⩽\frac{2d^{2}P_{d}^{2-1/d}}{n}, \end{array}$$




*and the relative error confidence as*

(10)
$$\begin{array}{c} P[|{\hat{P}}_{d}-P_{d}\left|>\epsilon P_{d}\right]⩽\frac{2d^{2}}{nP_{d}^{1/d}\epsilon^{2}}. \end{array}$$



**Remark** **1**(Efficient Implementation). *Birthday estimators can be efficiently computed, in memory O(n) and one-pass over samples, by using a dictionary to count empirical frequencies of observed elements. Such an implementation is given in the supplementary material [41].*

**Remark** **2**(Structural Assumptions). *The bounds from Theorem 1 use the statistic $P_{d}$ of the sampling distribution. This explicit dependency is beneficial, as further discussion clarifies. Lacking any prior knowledge, it can be estimated by the worst-case behavior, in terms of the alphabet size.*

**Algorithm 2:** HIGH-CONFIDENCE BIRTHDAY MOMENT ESTIMATOR**Data**: 
a moment order *d*samples $X_{1},\ldots ,X_{n}$, n⩾d, from an unknown distribution with p.m.f. $p_{X}$a confidence parameter δ
**Result**: an estimate $\hat{P_{d}}$ of $P_{d}=\sum_{x}p_{X}{\left(x\right)}^{d}$, the *d*-th moment of $p_{X}$

k←⌈8log(1/δ)⌉



ℓ←⌊n/k⌋

**for** 
j=1,…,k⌋ 
**do**
$|{\hat{P}}_{d}^{\left(j\right)}\leftarrow$
output of Algorithm 1
$X_{j\cdot \ell },\ldots ,X_{(j+1)\cdot \ell -1}$ on (j-th input block of length *ℓ*)
**end**


(


$\hat{P_{d}}\leftarrow \mathrm{MEDIAN}\left({\hat{P}}_{d}^{\left(j\right)},j=1,\ldots ,k\right)$



We see that the choice $n>6d^{2}\epsilon^{-2}/P_{d}^{1/d}$ guarantees $P[|{\hat{P}}_{d}-P_{d}|>\epsilon P_{d}]⩽\delta$ for $\delta =\frac{1}{3}$. Higher confidence (smaller δ) can be handled with the method of *Robust Mean Estimation*.

**Theorem** **2**(High-Confidence Moment Estimator). *For any ϵ>0, Algorithm 2 approximates $P_{d}$ with probability 1−δ and a relative error of ϵ provided that*
(11)$$\begin{array}{c} n⩾\left\lceil \frac{8d^{2}}{\epsilon^{2}P_{d}^{1/d}}\right\rceil \cdot \left\lceil 8\log\left(1/\delta \right)\right\rceil . \end{array}$$

**Remark** **3.**
*The constant can be refined a little based on the methods from [42].*


When comparing these results with prior work, we review the following aspects:1.**Novel techniques of broader interest**. We recall that analyzing variance formulas has been challenging for prior works on entropy estimation ([32,34] resorted to Poisson approximations, while [35] gave an involved combinatorial argument), even for the case d=2 (the lack of sharp analysis caused lots of difficulties in property testing [37]). As opposed to these ad hoc approaches, we establish the formula in a simple yet direct manner, pointing out that such formulas can be obtained by the techniques of *U-statistics*. When discussing applications, we will further benefit from these tools.2.**Clean and improved formulas.** Our confidence bound does not involve any implicit constants, while prior works in their main statements have unspecified dependencies on *d* (essentially, hiding more than absolute constants). We compare the accuracy bounds from this and prior works in Table 1 below. Our bound is strictly better given that $P_{d}$ is minimized at the uniform distribution and thus $P_{d}^{-1/d}⩽K^{-1+1/d}$ (with a large gap when the distribution is far from uniform), which establishes that the dependency on *d* is $4d^{2}$. Leveraging the theory of *U-statistics*, we will show that the factor $O(d^{2})$ is optimal, which is also a novel contribution.

### 3.2. Adaptive Estimation

As per our variance analysis, the performance actually depends not on the alphabet size *K* (that is ultimately the pessimistic bound) but rather on a more fine-grained statistic of *X*, namely $P_{d}$. Following the result in Theorem 2, we could hope for a moment estimation of
(12)$$\begin{array}{c} n{=}^{?}\Theta \left(\log\left(1/\delta \right)\right)\epsilon^{-2}d^{2}P_{d}^{-\frac{1}{d}}. \end{array}$$The obvious obstacle is that, in general, we do not know $P_{d}$ in advance. We solve this problem by developing an *adaptive algorithm*. It does not assume the right number of samples in advance but tries gradually and eventually terminates with high probability, giving the answer within the desired margin of error and using only a few more samples than the ideal bound conjectured above. Its core is a subroutine that guesses the moment value, gradually changing the candidate.

#### 3.2.1. Lower-Bounding Moments

The key ingredients of our approach are the following two subroutines: Algorithm 3 tests, based on samples, whether the moment is smaller or bigger than a proposed candidate; subsequently, Algorithm 4 loops the tester over a grid of candidate values.

**Algorithm 3:** MOMENTLESSTHAN**Data**: 
independent samples $X_{1},\ldots ,X_{n}{∼}^{iid}p_{X}$tested threshold *Q*access to an estimator $\hat{P}$ of $P_{d}$
**Result**: tests if $P_{d}⩽\frac{Q}{2}$ or $P_{d}⩾2Q$

$\hat{P}\leftarrow \hat{P}\;\left(X_{1},\ldots ,X_{n}\right)$

**if** 
$\hat{P}<Q$ 
**then**| 
**return** True**else if** 
$\hat{P}⩾Q$ 
**then**| 
**return**
False

**Algorithm 4:** MOMENTBOUND**Data**:
independent samples $X_{1},X_{2},\ldots {∼}^{iid}p_{X}$ (online access)access to Algorithm 3the sample size n(Q,δ) for Algorithm 3
**Result**: finds bound *Q* such that $\frac{Q}{2}⩽P_{d}⩽Q$

Q←1

**while***$Q⩾1/K^{d-1}\;\&\;b=\mathrm{True}$*  **do**

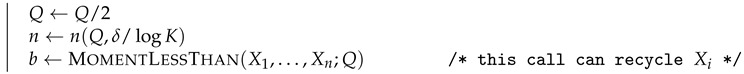


**end**
(**return** *Q*

The correctness of the approach is guaranteed by the lemmas stated below.

**Lemma** **1**(Moment Testing). *Let $\hat{P}$ be any estimator of $P_{d}$ which, given*
$$n⩾C\left(\delta \right)P_{d}^{-1/d}\epsilon^{-2}$$*samples, for some function C(·) and any ϵ>0,1>δ>0, achieves a relative error of ϵ with probability 1−δ. Then Algorithm 3, when given at least*
$$\begin{array}{c} n\left(Q,\delta \right)=\lceil 4C\left(\delta \right)Q^{-1/d}\rceil \end{array}$$*samples of X, with probability 1−δ, outputs* True *when
$P_{d}⩽Q/2$ and*
False
*when $P_{d}⩾2Q$.*

**Lemma** **2**(Moment Bounding). *With same $\hat{P}$ as in Lemma 1, with probability 1−δ, Algorithm 4 terminates after using at most*
$$\begin{array}{c} n=\lceil 4C\left(\delta /\left(d-1\right)\log K\right)P_{d}^{-1/d}\rceil \end{array}$$*samples of X, and its output Q satisfies $\frac{Q}{2}⩽P_{d}⩽4Q$.*

#### 3.2.2. Construction of Adaptive Estimator

Armed with Lemma 2, we are ready to analyze adaptive estimation. The algorithm is the same in the non-adaptive case, and we need Lemma 2 only to adjust the sample size.

**Theorem** **3**(Adaptive Moment Estimation). *With online access to samples from X, with probability 1−δ, one can estimate $P_{d}$ within a relative error of ϵ, terminating at the number of samples*
(13)$$\begin{array}{c} n⩽O(\left(\log\left(1/\delta \right)+\log\log K\right)P_{d}^{-\frac{1}{d}}\epsilon^{-2}). \end{array}$$

**Remark** **4**(Adaptive Overhead). *The sample size for adaptive estimation differs from the “dream bound” in (Equation 12) by a (very small) factor of loglogK.*

**Remark** **5**(Sample Complexity Guarantees). *Observe that the algorithm is not guaranteed to achieve the “good” sample complexity every time, but rather with high probability. This is a minor issue inherently related to the concept of adaptive estimation and does not affect much the performance in practical applications. Namely, we can always clip the total number of samples available at the pessimistic level from prior work and fall back to the fixed-size sample estimation should the adaptive estimation exceed the limit. This, however, happens with small probability δ, which can be further decreased with little overhead.*

**Remark** **6**(Comparison to Prior Work). *We give a clear comparison of our adaptive estimation and prior work in Table 2 below. We always have $P_{d}⩾Q$ and $P_{d}⩾K^{-1+1/d}$. Furthermore, usually $P_{d}/Q$ is much bigger than 1 (because in practice we do not know a prior lower bound in advance) and $P_{d}$ is much bigger than $K^{-1+1/d}$ (this gap can be as big as $K^{\Omega (1)}$, which is a huge factor for some applications; for example, in cryptography, we consider alphabets as big as $K=2^{256}$). Thus, our bound outperforms the previous approaches for typical use cases.*

### 3.3. Novel Applications

#### 3.3.1. One-Sided Estimation: Random Sources for Cryptography

The collision probability $P_{d}$ for d=2 plays an important role in cryptography: it quantifies the amount of randomness that can be extracted from a distribution [28]. For that extraction to work, $P_{2}$ should be small enough. Specifically, if $P_{2}⩽2^{-k}$ allows for extraction of nearly *k* bits of cryptographic quality, how could we check whether $P_{2}⩽2^{-k}$?

To solve the problem, we adapt Algorithm 4 by adding early stopping; namely, we quit the loop if $Q⩾2^{-k}$. We take $n=O(\log\left(k/\delta \right)2^{k/2})$ samples.

It remains for us to show that the algorithm behaves as desired. By the guarantees in Lemma 1 and the union bound over at most *k* steps, with probability 1−δ, we have the following: when $P_{2}<2^{-k-1}$, the algorithm finishes with $Q⩽2^{-k}$; and when $P_{2}>2^{-k+1}$, the algorithm finishes with $Q⩾2^{-k}$. This can be generalized to one-sided estimation for any *d*, where the goal is to decide whether $P_{d}>\Omega \left(2^{-k}\right)$ or $P_{d}<O\left(2^{-k}\right)$.

This one-sided estimation allows for saving samples and testing only up to a necessary extent. In cryptography, we do not have to estimate the whole entropy (which may be more costly, even with adaptive estimation) but only what suffices for the chosen application.

#### 3.3.2. Birthday Estimators Are UMVUE

The shortcut UMVUE stands for *uniformly minimum unbiased variance estimators*. We prove this conceptually strong and interesting characterization, which essentially shows that the birthday estimator in Theorem 1 is variance-optimal among unbiased estimators. The argument is inspired by our variance analysis, seeing the estimator as a *U-statistic*.

**Corollary** **1**(Birthday estimators are UMVUE). *Let ${\hat{P}}_{d}$ be as in Algorithm 1 and ${\tilde{P}}_{d}$ be another unbiased (for any X) estimator of $P_{d}$. Then we have $\mathrm{Var}\left[{\hat{P}}_{d}\right]⩽\mathrm{Var}\left[{\tilde{P}}_{d}\right]$.*

**Proof.** We will use the known result due to Lehmann and Sheff, which states that if an unbiased estimator is a function of a complete and sufficient data statistic, then it has the smallest possible variance [43,44].To apply this result, without losing generality (as it is a matter of encoding the alphabet), we assume that *X* takes values in the set {1,…,K}. Consider the sample $X_{1},\ldots ,X_{n}$, and let σ be the rearrangement such that $X_{\sigma (1)}⩽X_{\sigma (2)}⩽\ldots X_{\sigma (n)}$ (this is called the *order statistic*). The estimator ${\hat{P}}_{d}$ can be seen as the average of the symmetric function $h\left(X_{i_{1}},\ldots ,X_{i_{d}}\right)=I\left(X_{i_{1}}=x_{i_{2}}=\ldots =X_{i_{d}}\right)$ over tuples $i_{1}<i_{2}<\ldots <i_{d}$, and thus is also the function of $T=(X_{\sigma (1)},X_{\sigma (2)},\ldots ,X_{\sigma (n)})$. The claim follows if we prove that *T* is sufficient and complete (as a sample statistic).Order statistics are sufficient for univariate distributions. This is because we have:
$$\begin{array}{c} P_{X_{1},\ldots ,X_{n}|X_{\sigma (1)},X_{\sigma (2)},\ldots ,X_{\sigma (n)}}=\frac{1}{n!}, \end{array}$$
which does not depend on $X_{i}$. Thus, $X_{\sigma (1)},X_{\sigma (2)},\ldots ,X_{\sigma (n)}$ carries the same information about the data as $X_{i}$.The completeness of *T* means that there are no non-trivial unbiased estimators of zero; equivalently, if Ef(T)=0 for all sampling distributions and some function *f*, then P[f(T)=0]=1. To this end, observe that from the sufficiency proved above we have
$$\begin{array}{c} Ef\left(X_{\sigma (1)},\ldots ,X_{\sigma (n)}\right)=0⟹Ef\left(X_{1},\ldots ,X_{n}\right)=0 \end{array}$$Suppose that the above holds for any finitely supported sampling distribution *X*. Let *X* take values i∈I with probability $p_{i}$. Then the above implies
$$\begin{array}{c} \sum_{i_{1},\ldots ,i_{n}\in I\times \ldots \times I}p_{i_{1}}\ldots p_{i_{n}}f\left(i_{1},\ldots ,i_{n}\right)=0 \end{array}$$
for every distribution $\left(p_{i}\right)$. The left-hand side represents a multivariate polynomial in variable $p_{i}$, which evaluates to zero on the entire simplex of dimension n−1. Thus, its coefficients must be zero, which implies $f(i_{1},\ldots ,i_{n})=0$ for each tuple $i_{1},\ldots ,i_{n}$ and proves that *T* is sufficient. □

#### 3.3.3. Central Limit Theorem for Birthday Estimators

We again represent the estimator as the average sum over tuples:P^d=nd−1∑i1<…<idh(Xi1,…,Xid),
where
$$\begin{array}{c} h\left(x_{i_{1}},\ldots ,x_{i_{d}}\right)≜I\left(x_{i_{1}}=x_{i_{2}}=\ldots =x_{i_{d}}\right). \end{array}$$We view the whole expression as the *U-statistic* with the kernel function *h*. Then we show the following strong result (below, $N(0,\sigma^{2})$ denotes the normal distribution with zero-mean and variance $\sigma^{2}$).

**Corollary** **2**(Asymptotic Normality). *For n→+∞ it holds that $\sqrt{n}\cdot \left({\hat{P}}_{d}-P_{d}\right)\to N\left(0,\sigma^{2}\right)$ where $\sigma^{2}=d^{2}\left(P_{2d-1}-P_{d}^{2}\right)$, with $P_{d}$ as in* (Equation 3).

The proof utilizes the classical convergence results for *U-statistics* [38] and the derivation of our variance formula. Note that the result says that the central limit theorem applies, despite the fact that the sum components are correlated. Clearly, the result is interesting on its own, particularly because (a) it proves that our constant $O(d^{2})$ is sharp, and (b) can be used more generally to benchmark other proposed bounds, by means of comparing with the asymptotic gaussian tail.

However, we would like to point out an application to applied statistical research. In [36], Rényi entropy of order d=2 has been estimated for the distribution of physically unclonable functions (PUFs), which are important in the field of cryptography. However, their methodology lacks statistical rigor. Particularly, for the authors’ needs, prior work on Rényi entropy estimation was insufficient in terms of clarity on constants; thus, they resorted to the naive application of the central limit theorem, which can give very biased results.

A more solid alternative would be to use the above corollary to (a) justify the soundness, at least in the regime of large *n*, and (b) establish a more robust estimation of the variance.

**Proof** **of** **Corollary** **2.**The limiting variance equals $d^{2}\sigma_{1}^{2}$ [39] with
$$\begin{array}{c} \sigma_{1}^{2}=\mathrm{Cov}\left[h\left(X_{i_{1}},\ldots ,X_{i_{d}}\right),h\left(X_{j_{1}},\ldots ,X_{j_{d}}\right)\right], \end{array}$$
where the tuples $\left(i_{1},\ldots ,i_{d}\right)$ and $\left(j_{1},\ldots ,j_{d}\right)$ have only k=1 element in common. We analyze this expression when proving Theorem 1 and know that it equals $P_{2d-1}-P_{d}^{2}$. The claimed formula now follows. □

#### 3.3.4. Adaptive Testing in Evaluation of PUFs

Here we discuss again an application to [36], but from a different perspective. As explained by the authors, the problem with estimating Rényi entropy of PUFs is a serious bottleneck: for this problem, the alphabet is huge, which limits the experiment scope, even on computational clusters [36]. In this note, we would like to point out that parts of these difficulties can be solved by our adaptive estimation. In fact, PUFs provide an excellent use case when *entropy is quite low*; therefore, the moment $P_{d}$ term in Theorem 3 is much bigger than the pessimistic bound based on the alphabet size. We discuss this application in full detail in a follow-up work.

#### 3.3.5. Applications to Property Testing

The estimator from Algorithm 1 was first studied in [45], but the variance bounds obtained were not sharp. Quite oddly, in the ongoing research on closeness testing, the birthday-like collision estimators (being subroutines for uniformity checking) seemed to be suboptimal [46] until, very recently, the work of [37] re-examined the variance formula for d=2 and shows that it achieves (in our notation) optimal dependence on *K* and ϵ. Thus, a breakthrough was possible just because of a specialized version of (Equation 8). In this discussion, we would like to (a) point out that the general variance formula can likely have similar applications and impact for d>2 and should be of broader interest, and (b) comment on a minor gap in an early version of the proof of the central result in [37]. Lemma 2.3 in [37], which establishes the variance bound for d=2, is the key ingredient of the main results. The authors derive an expression bounding, in our notation, the variance in Theorem 1 for the case d=2. When doing so, they face up the term $n\left(n-1\right)\left(n-2\right)\left(P_{3}-P_{2}^{2}\right)$ (in our notation) and bound it as $n^{3}\left(P_{3}-P_{2}^{2}\right)$ (the last line of derivation claims the upper bound, the following remark claims the lower bound). The reasoning, however, is valid when $P_{3}-P_{2}^{2}$ is non-negative. This is true by Proposition 3.

### 3.4. Application to Statistical Inference

We will use Algorithm 1 to efficiently test whether a given distribution is close or far from a uniform one. The procedure described below is asymptotically equivalent but numerically superior to the one from [37].

Denote by *K* the alphabet size and let γ be such that $P_{2}=\frac{1}{K}+\gamma$. We see that
(14)$$\begin{array}{c} \gamma =\sum_{x}{\left(p_{X}\left(x\right)-\frac{1}{K}\right)}^{2}, \end{array}$$
which shows that γ measures the squared $\ell_{2}$-distance between $p_{X}$ and the uniform distribution. For convenience, we will refer to γ as the *collision gap*. Define
(15)$$\begin{array}{c} \hat{\gamma }=\hat{P_{2}}-\frac{1}{K}. \end{array}$$

This estimator gives an unbiased approximation of γ, because $E\left[\hat{\gamma }\right]=E\left[\hat{P_{2}}\right]-\frac{1}{K}=P_{2}-\frac{1}{K}=\gamma$. Furthermore, its variance equals the variance of $\hat{P_{2}}$ because $\frac{1}{K}$ is a deterministic constant. By Chebyshev’s inequality, with probability 1−δ, we have
(16)$$\begin{array}{c} |\hat{\gamma }-\gamma |⩽\sqrt{\mathrm{Var}[\hat{P_{2}}]/\delta }. \end{array}$$We now analyze the variance in more detail. Define $\delta \left(x\right)=p_{X}\left(x\right)-\frac{1}{K}$. Then $P_{2}=\frac{1}{K}+\sum_{x}\delta {\left(x\right)}^{2}$ and $P_{3}-P_{2}^{2}=\sum_{x}\delta {\left(x\right)}^{3}+\frac{1}{K}\sum_{x}\delta {\left(x\right)}^{2}-{\left(\sum_{x}\delta {\left(x\right)}^{2}\right)}^{2}$. In particular, we have $P_{2}=\frac{1}{K}+\gamma$ and $0⩽P_{3}-P_{2}^{2}⩽\gamma^{\frac{3}{2}}+\frac{\gamma }{K}$ by Propositions 3 and 4. Therefore, by Theorem 1 we obtain
(17)VarPd^−1K⩽4(γK+γ32)n+γ+1Kn2.Thus, with probability 1−δ, we have
(18)γ−4(γK+γ32)nδ+γ+1Kn2δ⩽γ^⩽γ+4(γK+γ32)nδ+γ+1Kn2δ.The above two-sided inequality allows us to estimate a range of possible values γ with respect to the (observed) statistics $\hat{\gamma }$, which yields high-confidence bounds for γ.

We illustrate the procedure numerically on real-world datasets. Data and results are summarized in Figure 1a,b. Our method confirms non-uniformity in both cases and provides confidence intervals. The details of the experiment are shared in the supplementary material [41].

### 3.5. Application to Entropy Estimation

The following experiment illustrates advantages of adaptive entropy estimation for distributions with large support and relatively low entropy, such as Zipf’s law.

Let *X* follow Zipf’s law with parameter s=1.1 and the support of $K=10^{4}$ elements. By numerical calculations, we find that $P_{2}\approx 0.40005$. Consider now the task of estimating entropy of *X* from samples. Theorem 3 allows us to save a large factor of about $K^{1/2}=10^{2}$ in the number of samples. Calculations show that on a sample of size about n=10,000, the algorithm from Algorithm 2 finds an approximation ${\hat{P}}_{d}=0.39898$ with a relative error $\epsilon =\frac{1}{2}$ and confidence 1−δ=0.95. The details appear in the supplementary material [41].

## 4. Conclusions

This work simplifies the variance analysis of collision estimators, establishing the closed-form exact formulas and improving upon prior data-oblivious bounds by making them dependent on certain data statistics. In particular, we use the derived formulas to estimate Rényi entropy adaptively, asymptotically, and give other applications.

Numerical experiments highlight the importance of the dependency of sample size on confidence. The constants involved exponentially affect the confidence, so that further improvements are of significance for many real-world inference problems. This problem is left for future research. 

## Figures and Tables

**Figure 1 entropy-25-00185-f001:**
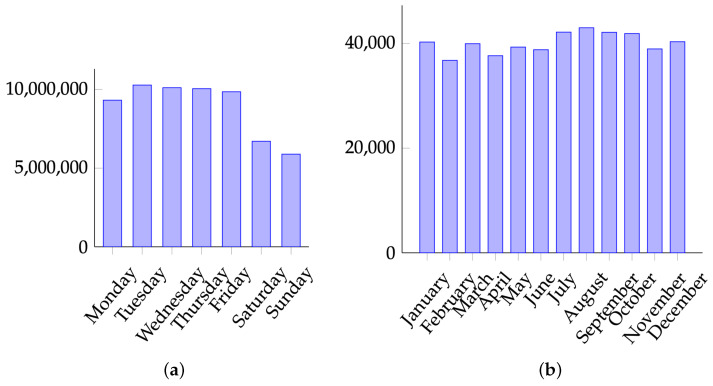
(**a**) U.S. births 2000–2014 (source: Social Security Administration). For this dataset, $\hat{P_{2}}\approx 0.14794$ and $\frac{1}{K}\approx 0.14286$, so that the gap equals γ≈0.00017. Our method gives the 99% confidence interval of (0.005,0.00516). (**b**) Births from insurance claims (source: courtesy of Roy Murphy). For this dataset, $\hat{P_{2}}\approx 0.08348$ and $\frac{1}{K}\approx 0.08333$, so that the collision gap equals γ≈0.00015. Our method gives the 99% confidence interval of (0.00009,0.00035).

**Table 1 entropy-25-00185-t001:** The performance of the “birthday estimator” of moments. In the formulas, ϵ∈(0,1) is the relative error, *n* is the sample size, and 1−δ is the confidence (the prob. that $|{\hat{P}}_{d}-P_{d}|⩽\epsilon P_{d}$).

Confidence 1−δ in Algorithm 1	Author	Assumption
$\delta ⩽4d^{2}n^{-1}\epsilon^{-2}P_{d}^{-1/d}$	**this paper**	n⩾d
$\delta ⩽O_{d}\left(n^{-1}\epsilon^{-2}P_{d}^{-1/d}\right)$	[35]	n⩾d
$\delta ⩽O_{d}\left(n^{-1}\epsilon^{-2}K^{1-1/d}\right)$	[32]	n⩾d

**Table 2 entropy-25-00185-t002:** The performance of moment (Rényi entropy) estimators. In the formulas, *K* is the alphabet size, ϵ is the relative error, and the confidence is 1−δ.

Sample Size *n*	Author	Assumptions
$O(\left(\log\left(1/\delta \right)+\log\log K\right)\epsilon^{-2}P_{d}^{-1/d})$	**this paper**	
$O(\log\left(1/\delta \right)\epsilon^{-2}Q^{-\frac{1}{d}})$	[35]	prior bound $Q⩽P_{d}$
$O(\log\left(1/\delta \right)\epsilon^{-2}K^{1-\frac{1}{d}})$	[32]	

## Data Availability

The data and code is shared in the GitHub repository [41].

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
