# Peer review of "Towards More Efficient Rényi Entropy Estimation"

_entropy, 2023, doi:10.3390/e25020185_

Round 1
Reviewer 1 Report
In this paper, the authors present a method to improve the estimation of Renyi entropy and propose a simpler analysis of the generalized birthday paradox collision estimator which outperforms the previous methods. I find this work interesting and clearly presented. To my knowledge, it is original and sound. So it can be published in Entropy.
Author Response
Dear Reviewer, thank you for the feedback.
I have improved the discussion of methods by adding numerical examples and sharing an interactive notebook (supplementary material). I hope this will improve the experience regarding "methods adequately described".
Reviewer 2 Report
The manuscript “Towards Better Rényi Entropy Estimation” presents variance analysis of collision estimators using an adaptive estimation of Rényi entropy, without prior information from sampling distribution. The manuscript is well written, with the results clearly explained, despite the lack of graphic illustration. The introduction presents the research topic in a broad context, highlighting various applications through references [2]-[28]. Although the manuscript deals with a subject that is worthy of investigation, it still needs to be improved. Please see some comments below.
> In line 92 the authors state that “Throughout the paper X is a fixed discrete distribution (...)”. Using X as a fixed discrete distribution is weird, especially when the sequence X1, X2, …, Xn is a set of random variables. X1, X2, ..., Xn implie that they are members of the random variable X. Please consider using X = {X1, X2, ..., Xn} and replacing X with another letter, such as P.
> Section 2.3. U-Statistics should be improved, even the authors should inform the meaning of the letter U, which, although it is well known, this approach may not be so obvious to many readers. Also, the collision estimators concepts should be clarified. These are crucial points to be addressed in this manuscript.
> Please, standardize the letter U of the term “U-statistics”. Also, double-check the journal names in references, some are abbreviated and others are not.
> The authors state that the "implementation is provided as supplementary material", but it has not been provided.
> Please include some perspectives in the section Conclusion.
Overall, the authors present the methodology and results accurately. In my opinion, the article should address the above points in order for the work to be interesting for several readers.
Author Response
Thank you for good suggestions!
Here is how I have addressed the comments:
- IID notation:
I follow the notation common in probability theory: some authors even say X_1,..,X_n are independent copies of X. This convention is used in top journals, for example I find it in "Corollary 1 in Ann. Probab. 25(3): 1502-1513 (July 1997). DOI: 10.1214/aop/1024404522".
But I acknowledge it may be confusing. I changed $X$ to $p_X$ - U-Statistics/Collisions:
Agree. More generally, the paper lacked elaborating on motivation around technical concepts and preliminaries. I extended the discussion on U-statistics and other parts. - Notation and references
I did efforts to improve the notation, and rewrote references so that they appear in the full format with DOI. - I added one important perspectives for conclusion: turns out that estimation still suffers from confidence boosting (despite much better constants elsewhere), which is a topic planned for future work.
- With regards to "methods adequately described" (a low grade) I hope that new applications and numerical examples backed by supplementary material on GitHub will improve the reader experience.
Reviewer 3 Report
The problem of estimation of functions of probability distributions is of current interest and this work has made contribution to provide new proofs and analysis.
My main concern is on the presentation. First, some proposition shall be proved in more detail, e.g., Prop. 2 or some background shall be given, e.g., Prop. 1 so that the contents are more self-contained. Second, the language needs more refinement. For example, line 2: or -> and; line 5: `than' shall be followed by 'that' or 'those'. Please go over the whole paper to improve the grammar.
Author Response
Dear Reviewer,
Thank you for the feedback. As you suggested, I elaborated on technical propositions to explain how they are used. I also improved the writing in many places.
Reviewer 4 Report
The manuscript “Towards Better Rényi Entropy Estimation” written by Skorsky
This author discussed improvement in Renyi entropy estimation.
This claim was provided straightforwardly.
Let us go step by step:
I) The primary results (as something new) are presented in Section 3; e.g., in Sec. 3.1 (“Algorithm”), "... is averaged over the d-element subsets of the sample". While there is better moment estimation claimed (yes, indeed), what is, on the other hand, a “dark side” of this approach in consideration? I mean, as far as computational costs needed for actual evaluation are concerned (e.g., in such a sampling), the costs may be higher than those paid in prior methodologies, especially if n is large enough. I'm wondering, accordingly, about the dark side of this one as trade-offs. Express it explicitly.
II) Section 3.3 Novel Applications. “Suppose that X is over an alphabet of size K = 2m, and that in addition we know an a-priori bound P_{max} = 2^{-\alpha m} for some \alpha < 1.” Through this statement, the parameter \alpha was introduced and used for exponential gain. How come with a justification of introducing \alpha? If this parameter is NOT available, this gain cannot be obtained at all, I guess. Clarify it.
III) Section 4 is used for proofs only. I would suggest this section to move to an Appendix, rather than staying in a regular section.
IV) Section 5: “The implementation is provided as supplementary material.” What is this material? I did not get it.
Consequently, I cannot recommend the publication of this manuscript in the present form.
Author Response
Thanks for the comments!
- Complexity the tuple aggregation indeed brings simplifications and accuracy, but also equally efficient: birthday estimators can be computed in linear memory and one pass over data, just as in previous approaches, using dictionaries to store frequencies. This is clarified in Remark after Theorem 1 and in examples given in supplementary material.
- I agree, that application was not specific enough. I rewrote this paragraph discussing one-sided estimation, which is more adequate and illustrative.
- I moved proofs to the appendix.
- The implementation is provided as an interactive notebook (Colab Jupyter), the link is in references.
- I expanded the application section and restructured preliminaries.
Round 2
Reviewer 4 Report
This manuscript is ready to publish in Entropy.